# Two-Step Energy Transfer Dynamics in Conjugated Polymer and Dye-Labeled Aptamer-Based Potassium Ion Detection Assay

**DOI:** 10.3390/polym11071206

**Published:** 2019-07-19

**Authors:** Inhong Kim, Ji-Eun Jung, Woojin Lee, Seongho Park, Heedae Kim, Young-Dahl Jho, Han Young Woo, Kwangseuk Kyhm

**Affiliations:** 1School of Electrical and Computer Science, Gwangju Institute of Science and Technology (GIST), Gwangju 61005, Korea; 2Department of Chemistry, Korea University, Seoul 02841, Korea; 3Department of Optics & Mechatronics Engineering, Pusan National University, Busan 46241, Korea; 4School of Physics, Northeast Normal University, Changchun 130024, China

**Keywords:** FRET, time-resolved photoluminescence, two-step FRET, potassium ion detection

## Abstract

We recently implemented highly sensitive detection systems for photo-sensitizing potassium ions (K^+^) based on two-step Förster resonance energy transfer (FRET). As a successive study for quantitative understanding of energy transfer processes in terms of the exciton population, we investigated the fluorescence decay dynamics in conjugated polymers and an aptamer-based 6-carboxyfluorescein (6-FAM)/6-carboxytetramethylrhodamine (TAMRA) complex. In the presence of K^+^ ions, the Guanine-rich aptamer enabled efficient two-step resonance energy transfer from conjugated polymers to dyed pairs of 6-FAM and TAMRA through the G-quadruplex phase. Although the fluorescence decay time of TAMRA barely changed, the fluorescence intensity was significantly increased. We also found that 6-FAM showed a decreased exciton population due the compensation of energy transfer to TAMRA by FRET from conjugated polymers, but a fluorescence quenching also occurred concomitantly. Consequently, the fluorescence intensity of TAMRA showed a 4-fold enhancement, where the initial transfer efficiency (~300%) rapidly saturated within ~0.5 ns and the plateau of transfer efficiency (~230%) remained afterward.

## 1. Introduction

In the context of various optical sensing applications, Förster resonance energy transfer (FRET) has been extensively investigated over decades due to its superior capacity to detect unknown particles and their conformational change at the molecular scale as well as their energy harvesting nature via amplification in the selected spectral windows [1,2,3,4,5,6]. FRET is a distance-dependent phenomenon, which is built on the basis of non-radiative energy transfer from energy donors to energy acceptors within close proximity (~10 nm) via long-range dipole–dipole interactions [7]. Thus, the design of FRET-based optical sensing assays always needs a platform for a proper intermolecular distance and a specificity for target molecules. Most of the myriad of recent FRET configurations are relevant to this intrinsic sensitivity to nanoscale change between two dipoles and the selection of proper materials, including fluorophores and recognition elements [8].

Conjugated polymers (CPs) have been utilized as an optical platform for many bio- or chemical applications due to their useful optical and electronic properties characterized by delocalized π-electrons [5,9,10,11,12,13,14,15,16,17,18]. In particular, cationic CPs with terminal quaternary ammonium groups were recently used in optical DNA sequence detection through electrostatic complexation, which provides a noble route to molecular distance control [1,2,5].

Aptamers are nucleic acid molecules that bind to specific targets, forming a secondary-folded structure [19,20]. Recently, they have attracted attention as an alternative conventional recognition component such as antibodies and various biosensor applications [21]. The advantages of aptamers compared with conventional recognition elements lies in their cost-effective production, easy modification, and low immunogenicity [22]. In particular, some specific single-stranded aptamers with guanine (G)-rich base sequences have a high affinity and high specificity for alkali metal ion. They can construct a secondary-folded structure, a so-called G-quadruplex, in the presence of specific alkali metal ions through hydrogen bonding [23,24].

As a one of the main cations in intracellular fluids in living bodies, potassium ion plays an important role in physiological activities as well as biological processes, for example, in maintenance of muscular strength, extracellular osmolality, enzyme activation, and apoptosis [25,26,27]. Because many diseases like diabetes, anorexia, bulimia, and heart disease are also closely related to abnormal potassium ion concentration, monitoring of potassium levels is crucial for clinical diagnosis [28]. Various studies for the detection of K^+^ ions have been reported; however, selectivity against other intra/extra-cellular cations (Na^+^) and detection sensitivity still need to be improved.

Recently, we demonstrated a noble potassium ion detection assay consisting of water-soluble CPs and dye-labeled aptamers based on FRET [1]. In this FRET system, dye-labeled aptamers play two roles simultaneously, as not only a scaffold for FRET signaling but also a receptor for metal ions. The presence of K^+^ ions within a solution results in the conformational change of complex molecules consisting of positively charged CPs and negatively charged aptamers. This phenomenon was observed through a dramatic fluorescence enhancement. Nevertheless, the dynamics of sequential energy transfers are completely unknown. 

When FRET is occurring, donor fluorophores absorb the energy under the irradiation of incident light, then transfer the excited energy to nearby acceptor materials. In the presence of proper acceptors, efficient energy transfer leads to significantly quenched donor fluorescence intensity, providing the amplified acceptor fluorescence. This intensity variation is often measured by time-integrated fluorescence measurement. However, the fluorescence intensity can easily vary due to the changes in intensity fluctuations of excitation light, photobleaching, and light scattering [29]. In particular, the presence of metallic particles can alter the surrounding conditions, which may influence the optical properties of molecules. They may also act as collisional quenchers of fluorescence [30]. Moreover, we have to separately distinguish complexation-induced quenching from FRET-based fluorescence signals to increase our understanding of the molecular dynamics. In general, the correlation between FRET efficiency and changes in donor lifetime can be supported by the equation below:
(1)$$\phi =1-\;\frac{\tau_{\mathrm{DA}}}{\tau_{D}}$$
where $\tau_{\mathrm{DA}}$ and $\tau_{D}$ are the fluorescence lifetimes of the FRET donor in the presence and absence of the FRET acceptor, respectively [31]. Since FRET efficiency is inversely proportional to the fluorescence lifetime of the donor fluorophore, the higher the FRET efficiency means, the shorter donor lifetime, suggesting a decrease in the excited lifetime of the donor is great evidence of FRET. Observing the time-related fluorescence of the FRET system will be helpful to optimize the condition for maximized FRET efficiency as well as to understand dynamic events involved in the intermolecular energy transfer phenomenon.

In this paper, we investigated the fluorescence decay dynamics of the conjugated polymer and aptamer-based 6-FAM/TAMRA complex. Following our previous demonstration of the two-step FRET-based K^+^ ion detection assay, we studied the dynamics of sequential energy transfer processes in terms of exciton population variation of FRET partners. When CPs were excited with 380 nm light, the population dynamics of CPs, 6-FAM, and TAMRA were compared in the absence and presence of K^+^ ions, respectively. Regarding the intermediate energy level of 6-FAM located in between the high-level CPs and the low-level TAMRA, we also excited 6-FAM selectively using 490 nm light. This enables the study of FRET from 6-FAM to TAMRA selectively. Those results allowed us to investigate the two different FRET processes separately, whereby the detection of K^+^ ions was evaluated quantitatively.

## 2. Materials and Methods 

### 2.1. Materials

All chemicals were purchased from Aldrich Chemical Co. (St. Louis, MA, USA) for measurement at room temperature. As a FRET donor, the polyfluorene-based CP was synthesized via the Suzuki coupling reaction of 2,7-bis(4,4,5,5-tetramethyl-1,3,2-dioxaborolan-2-yl)-9,9-bis(6′-bromohexyl)fluorene and 2,7-dibromo-9,9-bis(3,4-bis(2-(2-methoxyethoxy)ethoxy)phenyl)fluorene using (PPh_3_)_4_Pd(0) as a catalyst in toluene/water (2:1, volume ratio) at 85 °C for 36 h (yield: 66%), followed by successive quaternization with condensed trimethylamine at room temperature. As a FRET acceptor, 6-FAM and TAMRA dyes were used. Both dyes were labeled to a high-pressure liquid chromatography (HPLC)-purified molecular aptamer with 15 bases (5′-6-FAMGGTT GGTG TGGT TGG-6-TAMRA-3′) obtained from Sigma-Genosys (The Woodlands, TX, USA). For sensing of K^+^ ions, 20 μL of the aptamer stock solution was diluted to 2 mL buffer, and the resulting solution was incubated at 60 °C for 30 min with and without K^+^ ions. Detailed information about the sample fabrication including chemical synthesis and polymerization can be found elsewhere [1].

### 2.2. Time-Integrated Fluorescence and Quantum Yield Measurement

The fluorescence spectra of three fluorophores (CPs, 6-FAM, and TAMRA) dissolved in water solution were obtained by spectrofluorometer (Jasco, FP-6500, Hachioji, Tokyo, Japan), where a xenon lamp was used as an excitation source. The corresponding quantum yield of the CPs was estimated relative to a freshly prepared fluorescein solution in water at pH = 11.

### 2.3. Time-Resolved Fluorescence Measurement

The ultrafast decay dynamics of three fluorophores in the absence and presence of K^+^ ions were measured using a conventional time-correlated single photon counting (TCSPC) system (SPC-130EM, Becker & Hickl GmbH, Berlin, Germany). For excitation light sources, 380 and 490 nm were used, which were obtained by second harmonic generation from the fundamental laser light (680~1080 nm) of a femtosecond oscillator (Chameleon Ultra-II, Coherent, Santa Clara, CA, USA). The pulse duration and repetition rate of femtosecond light was 70 fs and 80 MHz, respectively. For the detection of time-resolved fluorescence, a high-speed single photon detector (PMH-100, Becker & Hickl GmbH) and a photomultiplier tube (PMT) were used. The temporal resolution was around 190 ps.

## 3. Results and Discussion

### 3.1. Two-Step FRET Process in CPs and Two-Dye-Labeled Aptamer Complex

Two-step FRET systems with three fluorophores have advantages compared to one-step FRET systems, such as efficiency enhancement through relay stations and better detection sensitivity due to the lower background fluorescence of acceptors [32,33,34]. We carefully designed the optical potassium detection system based on the sequential energy transfer. As the energy donors and acceptors for FRET, three fluorophores (CPs, 6-FAM, and TAMRA) were selected. The three fluorophores emit different colors (blue, green, and red). Based on Bazan’s scheme [2], which enables tuning of intermolecular distance at the molecular scale by Coulomb interaction between donor and acceptor fluorophores, CPs as FRET donors were positively charged due to their side chains, including cations. On the other hand, a guanine (G)-rich aptamer containing 15 bases (GGTT GGTG TGGT TGG) as a scaffold for FRET was negatively charged. Two dyes (6-FAM and TAMRA) were labeled to both ends of the aptamer. In particular, the molecular interaction between CPs and aptamer, such as electrostatic and hydrophobic interaction, can be controlled by varying the charge ratio of CPs and aptamer. For efficient attractive interaction (i.e., all aptamers were complexed with CPs), the concentration of CPs was adjusted by changing its repeating units on the backbone.

Figure 1 shows our potassium ion detection system schematically, which consists of CPs and the two-dye-labeled aptamer complex. Since both CPs and aptamer are flexible in water solvent, they will be randomly distributed in a solution after complexation. The complexation by the attractive interaction between CPs and aptamer keeps the two dyes (i.e., 6-FAM and TAMRA) in close vicinity of the CPs. While the excitons generated optically in the CPs will be diffused over the backbone of CPs due to its delocalized electronic nature, the excitation energy of CPs can be transferred to the two adjacent dyes of 6-FAM and TAMRA by dipole–dipole coupling. The dipole–dipole interaction depends on the dipole orientation factor ($\kappa^{2}$) [30], and $\kappa^{2}$ is calculated by the consideration of the spatial distribution of donor and acceptor dipoles and the freedom of dipole motion, which are limited by surrounding environment. Nevertheless, the allowed range of the dipole motion may be narrowly defined or close to isotropic [35]. In our FRET system, we assumed that $\kappa^{2}$ was 2/3 by the dynamic averaging of fluorophores [36].

In order to understand the energy migration mechanisms within a system, we assumed the simple case of a single donor–aptamer pair and the effective intermolecular distance for the energy transfer. The average separation between CPs and aptamer was in the 1~10 nm range. As shown in Figure 1a, the energy of CPs was transferred to both 6-FAM and TAMRA simultaneously. It is noticeable that the size of an aptamer is much longer than the effective FRET distance. Thus, only one-step energy transfer from CPs to 6-FAM or TAMRA is possible, i.e., CPs are a FRET donor while both 6-FAM and TAMRA will act as FRET acceptors. The energy transfer efficiency by FRET will be determined by the extent of the resonance coupling among each of the donor–acceptor pairs. However, FRET partners will also result in unwanted fluorescence quenching by the surrounding environment. The environmental factors, including electrostatic complex between charged molecules and physical response (hydrophobic or hydrophilic) of constituents over the solvent can affect fluorescence intensity. As a result, some of the energy can be dissipated through the exciton deactivation process, resulting in fluorescence quenching. In the case of organic molecules, various fluorescence quenchings often occur by molecular contact between the fluorophores and quenchers. Thus, if the intermolecular distance is sufficiently short and the energy levels of acceptors is low enough compared to that of donors, excitons can be separated into individual charges or the separated electrons and holes of donor can be transferred to acceptors by photo-induced charge transfer (PCT) [37]. When quenchers are randomly distributed, the fluorescence quenching near the radiation boundary is determined by the encounter distance and the quencher concentration [30]. Furthermore, delocalized excitons migrating along the CPs′ chains can also be scattered or dissociated into charge carriers by hole polarons or trap sites [38]. We have assumed that the intrinsic optical properties of constituents are unaffected or unchanged by the complexation and the addition of new molecules in order to simplify our understanding on phenomenological dynamics. Therefore, total energy transfer will be determined by the competition between energy harvesting one-step FRET and energy-wasting PCT because both processes have an intermolecular distance-dependent nature: FRET rate ($k_{\mathrm{FRET}}$) is inversely proportional to the sixth power of the donor–acceptor distance ($\sim 1/R_{\mathrm{DA}}^{6}$), while PCT rate ($k_{\mathrm{PCT}}$) has an exponential distance dependence ($k_{\mathrm{PCT}}\propto e^{-R_{\mathrm{DA}}}$) [39]. 

The presence of K^+^ ions leads to a new molecular configuration due to the specific binding between aptamers and metal ions, as shown in Figure 1b. This molecular configuration is known as g-quadruplex, with a planar motif generated from the pairing of four guanine residues through Hoogsteen-like hydrogen bonding [40,41]. Guanine (G)-rich base sequences of aptamers can make secondary-folded structures in the presence of metal ions. In practice, our guanine-rich aptamer is designed for this specific capturing of K^+^ ions, while there is no spectral shift in the intrinsic emission of 6-FAM and TAMRA [1]. In this case, one-step FRET is no more a dominant emission process. Since guanine tetrad (g-quartet) shortens the intermolecular distance among two dyes attached to both edges of an aptamer, both additional energy transfer between the two dyes and the sequential energy transfer from CPs to TAMRA via 6-FAM (two-step FRET) are possible. The 6-FAM will act as not only an intermediator for the two-step FRET, but also as a FRET acceptor in the one-step FRET. On the other hand, TAMRA always performs as a FRET acceptor due to its low bandgap energy. Consequently, TAMRA’s emission will be enhanced by CPs and 6-FAM. In other words, the fluorescence enhancement in TAMRA indicates the detection of K^+^ ions. The detailed molecular structures of CPs, 6-FAM, and TAMRA are given in Figure 1c.

### 3.2. The Spectral Overlap between Conjugated Polymer, 6-FAM, and TAMRA

Figure 2a–c shows the intrinsic optical properties of three fluorophores (i.e., CPs, 6-FAM, and TAMRA), respectively. Absorption spectra were shown in terms of the molar extinction coefficient and emission spectra were normalized by the maximum fluorescence intensity. The three fluorophores of CPs, 6-FAM, and TAMRA show a broad absorption and emission spectrum with a Stoke shift. Their mirror symmetric shape between absorption and emission spectra is attributed to the overlap of the initial and final wave functions relevant to the vibronic transition at thermal equilibrium [42], and the spectral intensity is determined by the probability amplitude of energy levels involved in the vibronic transition. From the absorption measurement, the molar extinction coefficient of CPs calculated by the Beer–Lambert law was 0.52 × 10^4^ M^−1^·cm^−1^ and CPs had an absorption maximum at 397.8 nm. Unlike absorption spectra, the blue color emission spectrum of CPs showed two peaks (i.e., 426.6 and 451.0 nm). The quantum efficiency of CPs was 0.58 in a water solvent. Both the spectrum of 6-FAM and TAMRA showed perfect mirror symmetry with high molar absorption coefficients (1.25 × 10^4^ M^−1^·cm^−1^ for 6-FAM and 1.96 × 10^4^ M^−1^·cm^−1^ for TAMRA). The maximum absorption peaks of two dyes appear at 490.4 and 567.6 nm, respectively. 6-FAM shows a greenish color emission with the peak wavelength at 514.4 nm. The TAMRA showed a reddish color emission with the emission maximum at 590.2 nm.

Since FRET phenomenon is analogous to the interaction of a coupled oscillation system, where two oscillators harmonically interact with each other at a separation R, the spectral overlap among FRET partners represents the extent of resonant coupling [43]. In FRET measurement, spectral overlap (*J*) can be simply calculated by integrating an overlap area between the normalized emission spectra of the donor $f_{D}\left(\lambda \right)$ and the absorption of the acceptor with a molar extinction coefficient ($\varepsilon_{A}\left(\lambda \right)$) [30]:
(2)$$J=\int f_{D}\left(\lambda \right)\varepsilon_{A}\left(\lambda \right)\lambda^{4}d\lambda$$

Figure 2d shows a spectral overlap in three fluorophores. The calculated spectral overlap between CPs and 6-FAM (2.47 × 10^−26^ M^−1^·nm^4^) was higher than that between CPs and TAMRA (8.00 × 10^−27^ M^−1^·nm^4^). This result implies that the excitation energy of CPs can be transferred effectively to 6-FAM compared to TAMRA when only one-step FRET process is considered in the absence of K^+^ ions. It is noticeable that the spectral overlap between 6-FAM and TAMRA (5.00 × 10^−26^ M^−1^·nm^4^) was higher than that involved with CPs. Furthermore, we can calculate a characteristic Förster distance ($R_{0}$) from the spectral overlap, where the FRET efficiency becomes 50% [8]. Regarding the three FRET partners, the three Förster distances were calculated for CPs/6-FAM (33.1 Å), CPs/TAMRA (27.4 Å), and 6-FAM/TAMRA (44.8 Å), respectively. These results imply that a selection of FRET partners and a system design is theoretically suitable for efficient FRET. Along with the spectral overlap, the Förster distance also provides a clue for the conformational change in molecular scale, and this can help us estimate a relative donor–acceptor distance through a theoretical FRET efficiency estimation [30,44]. However, the accuracy of the relative intermolecular distance calculation is limited due to the presence of various quenching processes because the original FRET equation assumes that the variation in energy transfer efficiency has only resulted in FRET. Although the intermolecular distance calculation for individual energy transfer processes is difficult due to the complexity of system, we can estimate total energy transfer efficiency regardless of the number of transfer steps.

### 3.3. The Fluorescence Decay Dynamics of the Conjugated Polymer, 6-FAM, and TAMRA

For a better understanding of the energy transfer processes among FRET partners in the absence and presence of K^+^ ions, we measured the time-resolved fluorescence. First of all, we measured the fluorescence decay of fluorophores before the complexation with the two-dye-labeled aptamer (denoted by “Free”) was measured individually to understand the intrinsic fluorescence decay time of fluorophores. Then, we compared it with the measured data in the absence and presence of K^+^ ions. The measured fluorescence decay curve was fitted by single- or multi-exponential decay function taking into account the change in the decay curvature [30]. In the fluorescence decay curve, the multiple decay components generally implied the contribution of additional decay pathways due to the molecular interaction to the whole fluorescence decay. Thus, we distinguished fast and slow decay components from the multi-exponential decay curvature by taking into account the intrinsic fluorescence decay time of fluorophores.

To calculate the energy transfer efficiency of FRET, we used a rate equation or its time-dependent differential equation. In general, the fluorescence decay rate ($k_{F}$) is defined by the sum of all radiative ($k_{\mathrm{rad}}$) and non-radiative decay ($k_{\mathrm{nonrad}}$) components [30].
(3)$$k_{F}=k_{\mathrm{rad}}+k_{\mathrm{nonrad}}=\frac{1}{\tau_{\mathrm{rad}}}+\frac{1}{\tau_{\mathrm{nonrad}}}$$
where $\tau_{\mathrm{rad}}$ and $\tau_{\mathrm{nonrad}}$ represent radiative and non-radiative decay time, respectively. Since the fluorescence intensity (I) is proportional to the exciton population (N), the time-dependent fluorescence decay can be characterized by a differential equation form of exponential function [39,45].
(4)I(t)=∫N(t)dt
(5)$$\frac{dN\left(t\right)}{dt}=g-\frac{N\left(t\right)}{\tau_{\mathrm{int}}}$$
where g indicates carrier (exciton) generation function by excitation light. $\tau_{\mathrm{int}}$ represents an intrinsic fluorescence decay time including all radiative and non-radiative decay components. When decay pathways of FRET and charge transfer are involved, the rate equation can be modified by an additional non-radiative decay time ($\tau_{nonrad}^{*}$) [45].
(6)$$\frac{dN\left(t\right)}{dt}=g-\frac{N\left(t\right)}{\tau_{\mathrm{int}}}-\frac{N\left(t\right)}{\tau_{nonrad}^{*}}$$

In particular, this differential rate equation effectively provides the relation between an individual fluorescence decay rate and a time-dependent exciton population. 

#### 3.3.1. The Fluorescence Decay of Conjugated Polymers

Figure 3 shows the fluorescence decay curve of CPs under 380 nm excitation where this wavelength corresponds to the absorption maximum of CPs and minimizes the direct absorption by 6-FAM. Free CPs before the complexation with aptamer show a single exponential decay curve with 0.38 ns (without K^+^) and 0.43 ns (with K^+^) decay time. In the absence of K^+^ ions, the fluorescence decay time of multi-exponential components in CPs were 0.35 ns (fast) and 1.76 ns (slow), respectively. In particular, the fluorescence decay curve of CPs with K^+^ rarely changed compared to that of Free CPs, and at initial decay time, one-step FRET occurs. This result represents that the complexation effect hardly changed the intrinsic fluorescence decay time of CPs. On the other hand, the addition of K^+^ ions significantly affected the fluorescence decay dynamics of CPs. Within 0.50 ns, the time-resolved fluorescence intensity of CPs showed a significant decrease, whereby two decay times of 0.19 ns (fast) and 1.46 ns (slow) were obtained. Regarding the FRET process occurring within several hundred ps time, we can infer that the energy transfer became effective due to the emergence of additional decay pathways relevant to the presence of K^+^ ions, i.e., two-step FRET.

Given the fluorescence decay rate equation including an additional exciton decay, we estimated the decay rate of CPs. The total rate equation and its variation (Δk(t)) before and after molecular interaction can be described as:
(7)$$k\left(t\right)=g-\left\{\frac{dN\left(t\right)}{dt}\right\}N{\left(t\right)}^{-1}$$
(8)$$\Delta k\left(t\right)=k_{i}\left(t\right)-k_{f}\left(t\right)$$
where subscripts denote initial (Free) and final states (complexation or after K^+^ ion addition). In the presence of K^+^ ions, the decay rate of CPs (3.58 ns^−1^) involved in the two-step FRET was the fastest compared to other cases (2.29 ns^–1^ for Free CPs and 2.58 ns^–1^ for complexation) at the start moment of fluorescence decay. Afterward, the decay rate of CPs with K^+^ ions drastically decreased while the decay rate of other cases barely changed until 1 ns. This result shows our prediction of an additional non-radiative decay pathway is plausible.

#### 3.3.2. The Fluorescence Decay of 6-FAM

The fluorescence decay dynamics of 6-FAM or TAMRA in the complexation phase of CPs and aptamer were also measured. The fluorescence decay of individual dyes (i.e., Free 6-FAM and Free TAMRA before complexation) was undistinguishable due to their low absorption coefficient at 380 nm excitation. First, we investigated the fluorescence decay dynamics of 6-FAM. The 6-FAM performs the role of an intermediator in the two-step FRET process in the presence of CPs, but it always becomes a FRET donor in the absence of CPs. To clarify the different roles of 6-FAM associated with CPs, we measured the fluorescence decay dynamics of 6-FAM in the absence and presence of K^+^ ions, as shown in Figure 4a. Two kinds of excitation lasers (i.e., 380 and 490 nm) were used. The 380 nm excitation excited carriers mostly in CPs, giving rise to a FRET from CPs to 6-FAM. On the other hand, 490 nm excitation generated carriers in 6-FAM selectively without any excitation in CPs. Hence, no FRET occurred from CPs to 6-FAM. Regardless of K^+^ ions, the fluorescence intensity of 6-FAM at the moment of excitation rarely changed under 380 nm excitation. However, under 490 nm excitation, the presence of K^+^ ions led to a significant decrease of the fluorescence intensity at 6-FAM. When FRET occurs, a decrease in the fluorescence intensity is generally observed in a donor molecule as evidence of the energy transfer. Thus, this decreased intensity at 6-FAM resulting from 490 nm excitation indicated a one-step FRET from 6-FAM to TAMRA, where 6-FAM was a donor in the FRET process. One should be reminded that no carrier was excited in CPS when 490 nm excitation was used. However, with 380 nm excitation, 6-FAM seemed to play a role as the two-step FRET intermediator. Initially, carriers were excited in CPs, then transferred to 6-FAM. Because the energy transfer from 6-FAM to TAMRA was efficient, the carriers in 6-FAM are immediately transferred to TAMRA. As a result, the population of 6-FAM remains constant due to the dynamic compensation effect. The intrinsic decay time of Free 6-FAM was 3.72 ns under 490 nm excitation. However, we observed multi-exponential decay in the absence of K^+^ ions. The two decay times were 0.36 ns (fast) and 2.09 ns (slow) at 490 nm excitation, respectively. The fast decay component also indicated FRET from 6-FAM to TAMRA. On the contrary, the fluorescence decay at 6-FAM with 380 nm excitation showed a single exponential shape with a longer decay time of 0.49 ns compared to that with 490 nm excitation. The elongated decay time possibly resulted from the population increase resulting from the energy transfer from CPs to 6-FAM.

Upon the addition of K^+^ ions, the fluorescence lifetime of 6-FAM at initial decay times decreased regardless of the excitation wavelengths. In the case of 490 nm excitation, the fast decay time was 0.25 ns and the slow decay time was 2.18 ns, respectively. With 380 nm excitation, the fluorescence decay time was 0.33 ns. Those results suggest the presence of K^+^ ions accelerated the fluorescence decay of 6-FAM through two-step FRET in addition to one-step processes. In particular, it was very noticeable that the fast decay component within 0.50 ns were identically observed at the different excitation wavelengths, while the decay curvatures barely changed afterward. This implies that each energy transfer pathways can be resolved in different timescales within temporal resolution of our experimental system. Nevertheless, the fluorescence intensity and decay curvature were determined by the combination of all decay pathways.

For a quantitative evaluation of the two-step FRET, we calculated the time-dependent energy transfer efficiency in the absence and the presence of K^+^ ions. From the change in time-resolved fluorescence intensity before and after K^+^ ion addition, the total energy transfer efficiency ($E_{\mathrm{total}}\left(t\right)$) of 6-FAM can be theoretically calculated by [44]:
(9)$$E_{\mathrm{total}}\left(t\right)=\frac{k_{\mathrm{FRET}}\left(t\right)}{k_{\mathrm{nonrad}}\left(t\right)+k_{\mathrm{FRET}}\left(t\right)}=\frac{I_{D}\left(t\right)-I_{\mathrm{DA}}\left(t\right)}{I_{D}\left(t\right)}$$
where $I_{D}\left(t\right)$ and $I_{\mathrm{DA}}\left(t\right)$ are the fluorescence intensity of a donor (D) in the absence and presence of an acceptor (A), respectively. Similar to the fluorescence decay of CPs, we found that the fast decrease of the energy transfer efficiency occurs within 0.5 ns regardless of the excitation wavelength as shown in Figure 4b. Afterward, the total transfer efficiency remained constant. Those results confirm the presence of a two-step energy transfer gain. In particular, the minus sign of the transfer efficiency indicating the reduction of the total energy was noticeable. Interestingly, we found that a loss of exciton population due to the energy transfer and the fluorescence quenching at 6-FAM was compensated by CPs.

#### 3.3.3. The Fluorescence Decay of TAMRA

Finally, we measured the fluorescence decay dynamics of TAMRA in the absence and presence of K^+^ ions, respectively. To discern individual energy migration by one- or two-step FRET from the total energy transfer process, the fluorescence decay of TAMRA without 6-FAM (single dye-labeled aptamer denoted by “CP/TAMRA”) was also measured. Figure 5a shows the fluorescence decay of TAMRA under 380 nm laser excitation. As discussed above, the addition of K^+^ ions resulted in an enhancement of the acceptor′s exciton population by not only an additional energy transfer but also the fundamental interaction of the complexation with CPs. Consequently, in the presence of K^+^ ions, the fluorescence intensity of TAMRA significantly increased compared to that of the fundamental one-step energy transfer. The four-folded enhancement of fluorescence intensity can be understood as a result of the energy transfer since TAMRA is always an energy acceptor in our FRET system. Interestingly, TAMRA without 6-FAM had the highest fluorescence intensity at the moment of excitation despite their weak absorption coefficient at 380 nm and the narrow spectral overlap. Its fluorescence intensity was much higher than that of others. This unusual feature may be attributed to the number of FRET participants regarding the total amount of energy CPs offered was fixed. In the case that there is no 6-FAM, the energy exchange with TAMRA is the only possible pathway. But, the preferential direction of the energy migration is decided by the spatial distance from CPs to dyes as well as the probability wavefunction distribution of the two dyes in the solution. Assuming that the total amount of energy transferred from CPs is equal to the fluorescence intensity of TAMRA without 6-FAM, the fluorescence intensity ratio at the moment of the excitation are $I_{\mathrm{FT}}/I_{T\;\mathrm{only}}\approx 0.19$ without K^+^ ions and $I_{\mathrm{FT}}/I_{T\;\mathrm{only}}\approx 0.64$ with K^+^ ions, where $I_{T\;\mathrm{only}}$ and $I_{\mathrm{FT}}$ indicate the fluorescence intensities of TAMRA-labeled and two-dye-labeled aptamers, respectively. Since the emission of TAMRA is irrelevant to that of 6-FAM in the absence of K^+^ ions, we can predict the probabilistic apportionment of the excitation energy. On the other hand, as expected by the Förster distance, this result may indicate that one-step FRET from CPs to TAMRA is preferred compared to 6-FAM in spite of a weak absorption and narrow spectral overlap. Since the size of the aptamer is smaller than that of polymers, CPs may surround the aptamer within the system and the average separation of TAMRA from CPs may be closer. Considering that the quenching effect was not observed in TAMRA unlike 6-FAM, this assumption may be reasonable. 

Despite the fluorescence enhancement in the presence of K^+^ ions, the fluorescence decay curvature (~1 ns decay time) of TAMRA barely changed due to the non-radiative transfer nature of the FRET, as shown in the inset of Figure 5a. Furthermore, we estimated the energy transfer efficiency (η(t)) of TAMRA described in Reference [44]:
(10)$$\eta \left(t\right)=\frac{\int I_{{\mathrm{with}\;K}^{+}}\left(t\right)dt-\int I_{w/{o\;K}^{+}}\left(t\right)dt}{\int I_{w/{o\;K}^{+}}\left(t\right)dt}$$
where $I_{w/{o\;K}^{+}}\left(t\right)$ and $I_{{\mathrm{with}\;K}^{+}}\left(t\right)$ indicate the fluorescence intensity of TAMRA at time t in the absence and presence of K^+^ ions, respectively. This equation was described in the view of the gain or loss of the exciton population during the two-step FRET process. In the absence of K^+^ ions, the population of TAMRA was supported by CPs through one-step FRET. The fluorescence intensity of TAMRA in the absence of K^+^ ions was 100 at 2 ns in arbitrary units. On the other hand, if K^+^ ions were involved, an additional two-step FRET occurred from CPs to TAMRA via 6-FAM. The TAMRA plays an important role as an intermediator, where the energy transfer from CPs to TAMRA is mediated. The energy transfer efficiency equation enables to estimate the transferred fluorescence from TAMRA in terms of the normalized ratio to the fluorescence intensity of TAMRA in the absence of K^+^ ions. When the increased fluorescence intensity of TAMRA at 2 ns was 330, the energy transfer efficiency of 230% was obtained. It is surprising that two-step FRET resulted in a remarkable population enhancement. Such a large enhancement can be attributed to the delocalized electronic nature of CPs, giving rise to the energy harvesting effect.

In summary, Table 1 shows the fitted parameters for the fluorescence decay of CPs, 6-FAM, and TAMRA. The fluorescence decay time is calculated by single- or multi-exponential fitting, as given below [30]:(11)$$I\left(t\right)=I_{0}+a_{1}\exp\left(-\frac{t}{\tau_{1}}\right)+a_{2}\exp\left(-\frac{t}{\tau_{2}}\right)$$
where $a_{1}$ and $a_{2}$ are the weight factors of each of the decay components. $\tau_{1}$ and $\tau_{2}$ are decay time. In the presence of additional decay pathways due to the conformational change or molecular interaction, the fluorescence decay time decrease and its curvature can be changed from single exponential to multi-exponential shapes as well as intensity variation. Before and after complexation, this phenomenon representing occurrence of one- or two-step FRET was commonly observed in CPs and 6-FAM. In the case of TAMRA, the reduction in decay time may result in a change in the surrounding environment. In the presence of K^+^ ions, the two-step FRET process can be accelerated owing to a shortened intermolecular distance and, therefore, fluorescence decay time decreases more.

## 4. Conclusions

We investigated the ultrafast fluorescence decay dynamics in CPs and the aptamer-based 6-FAM/TAMRA complex. In this FRET system, the conformational change by K^+^ ions promoted the sequential energy transfer from CPs, through 6-FAM, to TAMRA, as well as the fundamental one-step FRET due to the complexation. To understand the energy migration mechanisms within the system, we phenomenologically analyzed and modelled in terms of the exciton population. Also, we experimentally observed the evidence of two-step FRET and the role of three fluorophores thorough the time-resolved spectroscopic technique. As we expected, the presence of the two-step FRET process due to the presence of the K^+^ ion was identified by the fluorescence enhancement (4 folded) and the change in the decay curvature. In particular, we found the role of the transition of 6-FAM as an intermediator as well as the energy receiver during the energy transfer process. To distinguish the individual energy transfer step, we also calculated the energy transfer efficiency by the rate equation. Our transfer efficiency calculation, relevant only to total energy variation, minimizes the overestimation due to the theoretical FRET rate calculus. Consequently, a steady energy transfer efficiency (230%) was observed within the period of radiative decay time.

## Figures and Tables

**Figure 1 polymers-11-01206-f001:**
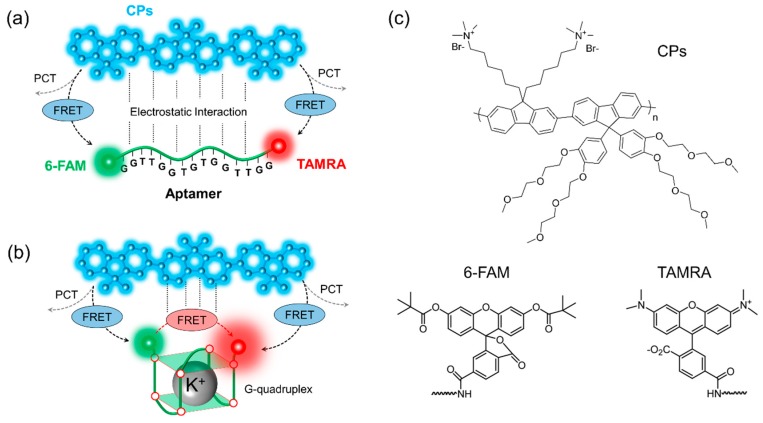
Schematics for potassium detection based on FRET. (**a**) In the absence of K^+^ ions, the dominant energy transfer process occurs from CPs to 6-FAM or TAMRA, that is, one-step FRET; (**b**) In the presence of K^+^ ions, by preferred molecular interaction between metal ions and guanine bases, the secondary structure (g-quadruplex) leads to the sequential energy transfer from CPs to 6-FAM to TAMRA, that is, two-step FRET; (**c**) The chemical structure of CPs, 6-FAM, and TAMRA.

**Figure 2 polymers-11-01206-f002:**
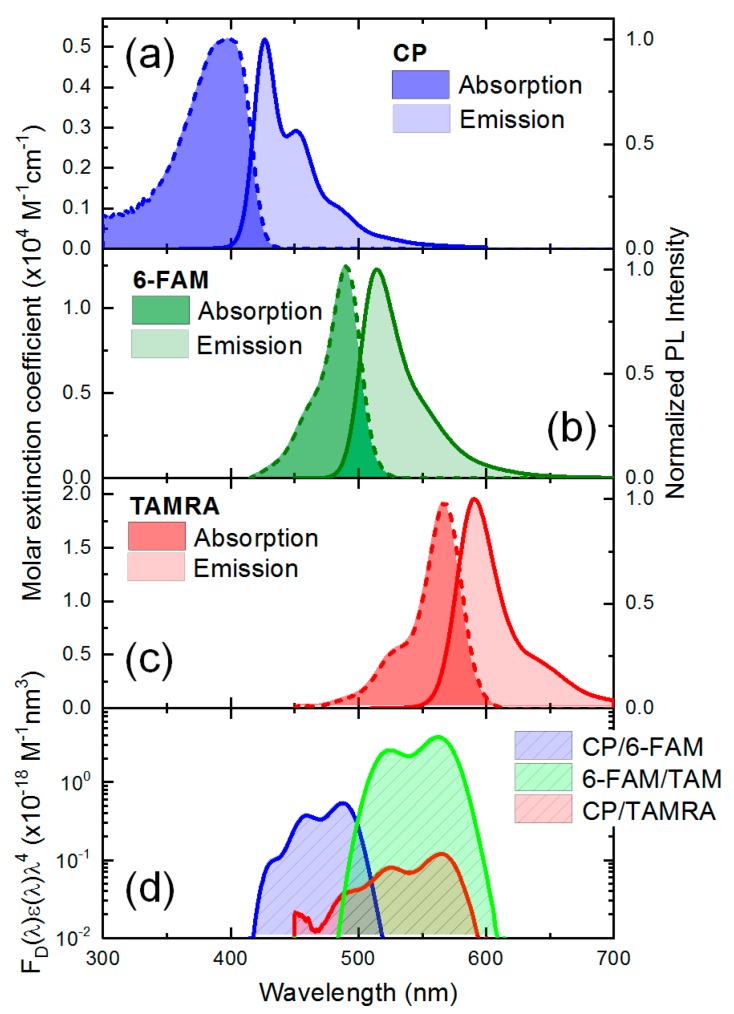
The optical properties of the three fluorophores (i.e., CPs, 6-FAM, and TAMRA) and the spectral overlap among them. The absorption and the emission spectra of (**a**) CPs; (**b**) 6-FAM; and (**c**) TAMRA were characterized in terms of the molar extinction coefficient and normalized by the maximum fluorescence intensity, respectively; (**d**) The spectral overlap between the three fluorophores was calculated by integrating the product of the absorption spectra and the emission spectra.

**Figure 3 polymers-11-01206-f003:**
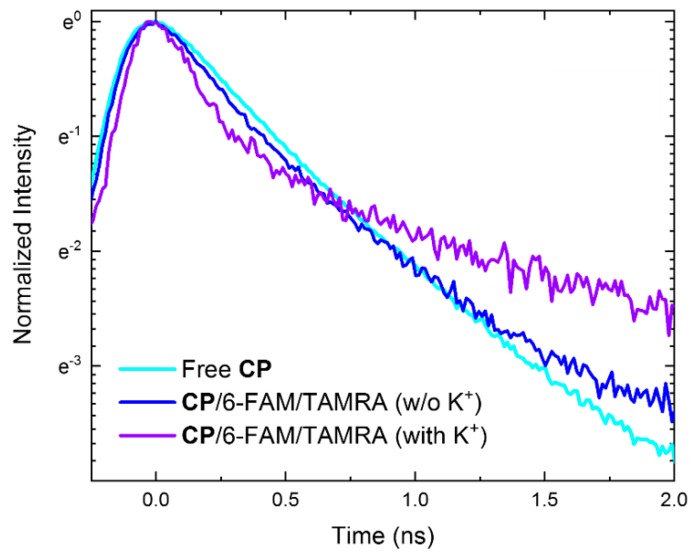
Fluorescence decay dynamics of CPs without the complexation with the two-dye-labeled (i.e., 6-FAM and TAMRA) aptamer (Free CP), in the absence (*w*/*o*) and presence (with) of K^+^ ions after the complexation with the two-dye-labeled aptamer.

**Figure 4 polymers-11-01206-f004:**
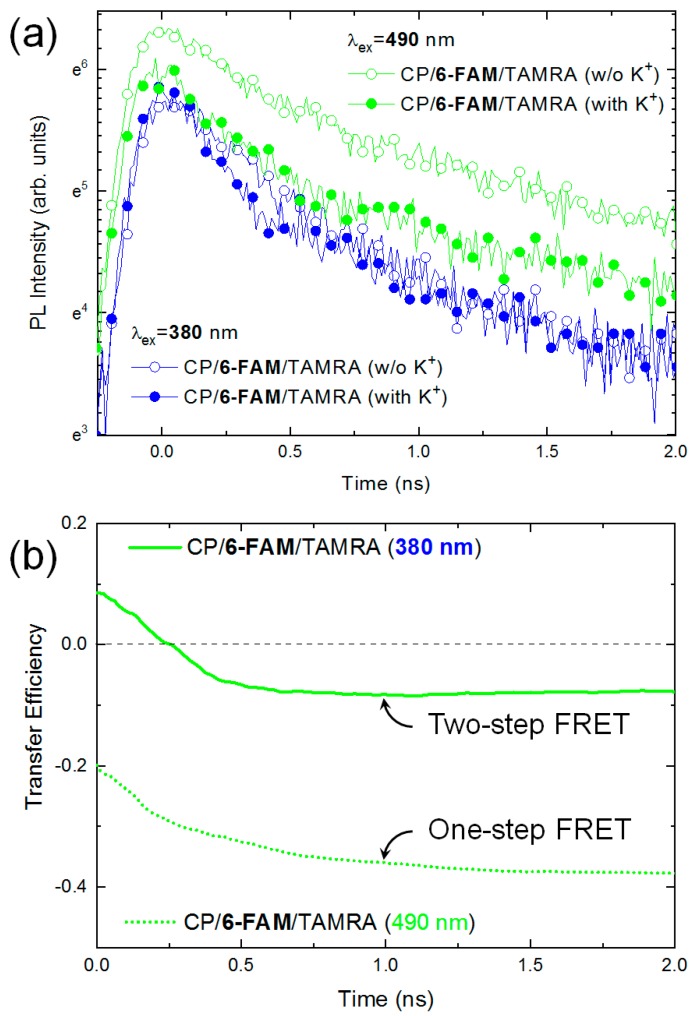
(**a**) The fluorescence decay of 6-FAM in the absence and presence of K^+^ ions. Time-resolved fluorescence decay was measured by two kinds of excitation wavelength (**blue** for 380 nm and **green** for 490 nm); (**b**) The energy transfer efficiency of 6-FAM for one-step FRET (**dotted line** for 490 nm excitation) and two-step FRET (**solid line** for 380 nm excitation) was calculated using Equation (9), respectively.

**Figure 5 polymers-11-01206-f005:**
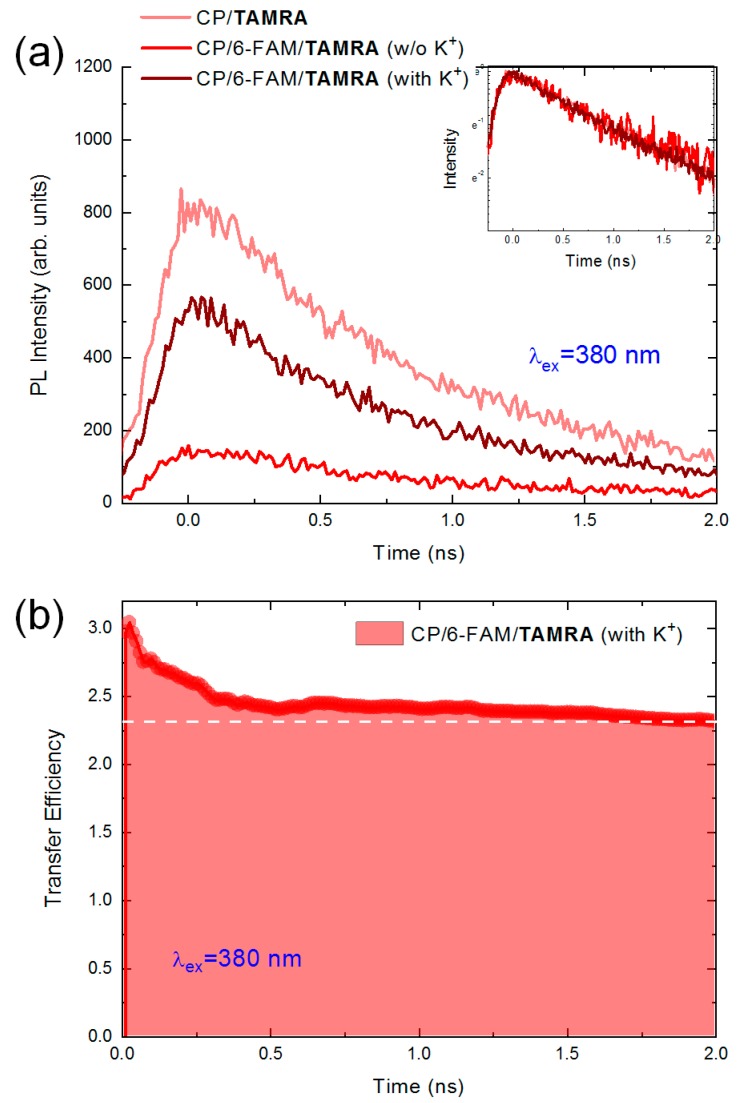
(**a**) To confirm the energy transfer from CPs to TAMRA, the fluorescence decay of TAMRA without 6-FAM was measured (**light red**, CP/TAMRA). The fluorescence decay of TAMRA in the absence (**red**) and presence (**wine**) of K^+^ ions; (**b**) The total transfer efficiency TAMRA gained by two-step FRET was calculated using Equation (10).

**Table 1 polymers-11-01206-t001:** The fitted parameters for the fluorescence decay of CPs, 6-FAM, and TAMRA.

Fluorophores	$\lambda_{ex}$	Free	*w*/*o* K^+^				with K^+^			
	(nm)	(ns)		(ns)		(ns)		(ns)		(ns)
		$\tau_{1}$	$a_{1}$	$\tau_{1}$	$a_{2}$	$\tau_{2}$	$a_{1}$	$\tau_{1}$	$a_{2}$	$\tau_{2}$
CPs	380	0.43 (with K^+^)	0.94	0.35	0.06	1.76	0.73	0.19	0.27	1.46
6-FAM	380	-	-	0.49	-	-	-	0.33	-	-
	490	3.72	0.51	0.36	0.49	2.09	0.61	0.25	0.39	2.18
TAMRA	380	1.01 (with CPs)	-	0.87	-	-	-	0.90	-	-

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
