# Peer review of "Two-Step Energy Transfer Dynamics in Conjugated Polymer and Dye-Labeled Aptamer-Based Potassium Ion Detection Assay"

_polymers, 2019, doi:10.3390/polym11071206_

Reviewer 1 Report

 In this paper, the authors investigated the ultrafast fluorescence decay dynamics in CPs and aptamer-based 6-FAM/TAMRA complex. In this FRET system, the conformational change by K+ ions promote the sequential energy transfer from CPs, through 6-FAM, to TAMRA as well as the fundamental one-step FRET due to the complexation.

Based on the above fundamental, the authors claimed to build the dye-labeled aptamer-based potassium ion detection assay. However, in contrast to K+, Pb2+ has a higher G-quadruplex stabilizing efficiency resulted from the formation of a more compact structure. For this potassium ion detection assay, the selectivity needs to be further evaluated against other common metal ions including Pb2+, Na+, K+, Zn2+, Ni2+, Fe3+, Co2+, Al3+, and Mg2+ ions. In addition, it is suggested that some references on the design of conjugated polymers (e.g, Sol. Energ. Mat. Sol. C, 2019, 103; J. Mater. Chem. A, 2016, 17604; J. Phys. Chem. Lett. 2018, 9, 6955; ChemSusChem, 2018, 360; ACS Appl. Mater. Inter., 2018, 762) could be cited.

Thus, I suggest the acceptance of this paper after major revision.

Author Response

Point 1: In this paper, the authors investigated the ultrafast fluorescence decay dynamics in CPs and aptamer-based 6-FAM/TAMRA complex. In this FRET system, the conformational change by K+ions promote the sequential energy transfer from CPs, through 6-FAM, to TAMRA as well as the fundamental one-step FRET due to the complexation.

Based on the above fundamental, the authors claimed to build the dye-labeled aptamer-based potassium ion detection assay. However, in contrast to K+, Pb2+ has a higher G-quadruplex stabilizing efficiency resulted from the formation of a more compact structure. For this potassium ion detection assay, the selectivity needs to be further evaluated against other common metal ions including Pb2+, Na+, K+, Zn2+, Ni2+, Fe3+, Co2+, Al3+, and Mg2+ ions. In addition, it is suggested that some references on the design of conjugated polymers (e.g, Sol. Energ. Mat. Sol. C, 2019, 103; J. Mater. Chem. A, 2016, 17604; J. Phys. Chem. Lett. 2018, 9, 6955; ChemSusChem, 2018, 360; ACS Appl. Mater. Inter., 2018, 762) could be cited.

Thus, I suggest the acceptance of this paper after major revision.

Response 1: In our former work (Adv. Funct. Mater. 2014, 24, 1748–1757), we have already investigated the selectivity of this sensory system against a range of other metal ions. The two-step FRET-induced PL signal was measured in the presence of various metal ions 20 mM NaCl, CaCl2, LiCl, MgCl2, NH4Cl, CuCl2, AgCl, ZnCl2, and AlCl3 by the same method used for K+ ion detection. As shown in below S-Figure 1 (a), the two-step FRET-induced selectivity of conjugated polymers (CPs)/aptamer against various metal ions is given as the ratio of fluorescence intensity. I425 and I585 represent the fluorescence intensity of CPs and TAMRA at the maximum emission wavelength, respectively. In the presence of potassium ions (K+), TAMRA shows strong FRET-induced fluorescence, i.e. CPs/aptamer system has a high specific binding affinity. Furthermore, their selectivity for K+ ions was not affected by the existence of other metal ions as shown in S-Figure 1 (b).

We added more references you suggested and revised manuscript.

à Revised sentences (Page 1): Conjugated polymers (CPs) have been utilized as an optical platform for many bio- or chemical applications due to their useful optical and electronic properties characterized by delocalized π-electrons [5,9-18]. In particular, ~

As Reviewer suggested, additional references were included:

14.  Liu, Z.; Zhang, L.; Shao, M.; Wu, Y.; Zeng, D.; Cai, X.; Duan, J.; Zhang, X.; Gao, X. Fine-tuning the quasi-3D geometry: Enabling efficient nonfullerene organic solar cells based on perylene diimides. ACS Appl. Mater. Interfaces 2018, 10, 762−768.

15.  Wen, S.; Wu, Y.; Wang, Y.; Li, Y.; Liu, L.; Jiang, H.; Liu, Z.; Yang, R. Pyran-bridged indacenodithiophene as a building block for constructing efficient A-D-A-type nonfullerene acceptors for polymer solar cells. ChemSusChem 2018, 11, 360 – 366.

15.  Wen, S.; Wu, Y.; Wang, Y.; Li, Y.; Liu, L.; Jiang, H.; Liu, Z.; Yang, R. Pyran-bridged indacenodithiophene as a building block for constructing efficient A-D-A-type nonfullerene acceptors for polymer solar cells. ChemSusChem 2018, 11, 360 – 366.

16.  Liu, Z.; Wu, Y.; Zhang, Q.; Gao, X. Non-fullerene small molecule acceptors based on perylene diimides. J. Mater. Chem. A, 2016, 4, 17604–17622.

17.  Liu, Z.; Gao, Y.; Dong, J.; Yang, M.; Liu, M.; Zhang, Y.; Wen, J.; Ma, H.; Gao, X.; Chen, W.; Shao, M. Chlorinated wide-bandgap donor polymer enabling annealing free nonfullerene solar cells with the efficiency of 11.5%. J. Phys. Chem. Lett. 2018, 9, 6955−6962.

18.  Liu, Z.; Zeng, D.; Gao, X.; Li, P.; Zhang, Q.; Peng, X. Non-fullerene polymer acceptors based on perylene diimides in all-polymer solar cells. Sol. Energ. Mat. Sol. C. 2019, 189, 103–117.

Reviewer 2 Report

In this manuscript Kim et Al. report an investigation of energy transfer process using fluorescence decay dynamics that occour between a conjugated polymer and aptamer-based 6-FAM and TAMRA fluorophores.

In principle the work is well done and to improve this work I could only suggest some marginal corrections (see below).
However, in my opinion, the manuscript is not suitable for POLYMERS as it is very specialized and better suited to a journal of physical chemistry.
From the point of view of the material and its application there is no novelty and the only improvement respect to the work previously published by the authors is based on the time-resolved spectroscopy.

Minor changes:
-Specify the acronyms at first use (eg CPs)
-Use the apex for the positive sign near potassium.
-Insert the 6-FAM and TAMRA structures to facilitate the reader in identifying the chromophore.
.Enter the correct polymer structure. The one represented in figure 1 is a simplification (already used in their previous work, however in the previous work the synthetic scheme and the correct structure is also reported).
-Add a motivation and possible future applications for a potassium ion detector.

Author Response

Response to Reviewer 2 Comments

Point 1: In this manuscript Kim et Al. report an investigation of energy transfer process using fluorescence decay dynamics that occur between a conjugated polymer and aptamer-based 6-FAM and TAMRA fluorophores. 

In principle the work is well done and to improve this work I could only suggest some marginal corrections (see below).
However, in my opinion, the manuscript is not suitable for POLYMERS as it is very specialized and better suited to a journal of physical chemistry.
From the point of view of the material and its application there is no novelty and the only improvement respect to the work previously published by the authors is based on the time-resolved spectroscopy.

Minor changes:
-Specify the acronyms at first use (eg CPs)
-Use the apex for the positive sign near potassium.
-Insert the 6-FAM and TAMRA structures to facilitate the reader in identifying the chromophore. Enter the correct polymer structure. The one represented in figure 1 is a simplification (already used in their previous work, however in the previous work the synthetic scheme and the correct structure is also reported).
-Add a motivation and possible future applications for a potassium ion detector.

Response 1: We revised our manuscript as Reviewer advised.

- First, the abbreviation of FRET and CPs are specified at first use.

à In the context of a various optical sensing applications, Förster resonance energy transfer (FRET) has been extensively investigated ~.

Conjugated polymers (CPs) have been utilized as an optical platform ~.

- Second, we corrected all positive sign of potassium ion.

à e.g. ~ potassium ion (K+) based on the two-step Förster resonance energy transfer.

- Third, we added the chemical structure of CPs, 6-FAM, and TAMRA to figure 1. And we revised manuscript and figure caption.

à  Revised sentences (Page 4): in Figure 1 caption, ~ (c) The chemical structure of CPs, 6-FAM, and TAMRA.

 in text, ~ The molecular structures of three fluorophores are described in Figure 1 (c).

- Finally, we add a motivation and possible future applications for a potassium ion detector to ‘Introduction’ and revised manuscript according to your comments.

à Revised sentences (Page 2): ~ As one of main cation in intracellular fluids in living bodies, potassium ion acts important role in physiological activities as well as biological processes, such as maintenance of muscular strength, extracellular osmolality, enzyme activation, and apoptosis [25-27]. Because many diseases like diabetes, anorexia, bulimia, and heart disease were also closely related to the abnormal potassium ion concentration, monitoring of potassium level is crucial approach for clinical diagnosis [28]. Various studies for the detection of K+ ions have been reported but selectivity against other intra-/extracelluar cations (Na+) and detection sensitivity are still need to be improved ~.

Reviewer 3 Report

The manuscript entitled “Two-step Energy Transfer Dynamics in Conjugated Polymer and Dye-labeled Aptamer-based Potassium Ion Detection Assay” focuses on the time-resolved fluorescence measurement of the studied system. This is straight continuation of research published in Adv. Funct. Mater. 2014, 24, 1748-1757. In the present form the Authors do not show the need for such research. Despite this, the manuscript contains elements of novelty. Taking this into account, the manuscript requires thorough improvement.

1.   The Introduction section does not indicate the need for conducted research. The authors should pay more attention to explaining the role of the time-resolved fluorescence measurement in similar studies.

2.    The structure of FAM, TAMRA, CP should be clearly shown.

3.   In the present form the text in the subsection 3. “Results and Discussion” is incomprehensible in fragments. The text should be improved and divided into more paragraphs and/or subsections.

4.   The decay time of fluorescence components should be summarized in the form of table or added to the Figure 4.

5.   A steady energy transfer efficiency was estimated as 230 %. The Authors should explain a result above 100%.

6.   Theoretical parts and discussion  should be supported by references

7.   Some editing issues:

- Paga 10 line 324

- Page 8 line 286

Author Response

The manuscript entitled “Two-step Energy Transfer Dynamics in Conjugated Polymer and Dye-labeled Aptamer-based Potassium Ion Detection Assay” focuses on the time-resolved fluorescence measurement of the studied system. This is straight continuation of research published in Adv. Funct. Mater. 2014, 24, 1748-1757. In the present form the Authors do not show the need for such research. Despite this, the manuscript contains elements of novelty. Taking this into account, the manuscript requires thorough improvement.

Point 1: The Introduction section does not indicate the need for conducted research. The authors should pay more attention to explaining the role of the time-resolved fluorescence measurement in similar studies.

Response 1: As Reviewer advised, we carelly revised ‘Introduction’ to explain the role of the time-resolved fluorescence measurement.

à Revised sentences (Page 2): ~ When FRET is occurring, donor fluorophores absorb the energy under the irradiation of incident light then transfer the excited energy to nearby acceptor materials. In the presence of proper acceptor, efficient energy transfer lead to significantly quenched donor fluorescence intensity, providing the amplified acceptor fluorescence. This intensity variation is often measured by time-integrated fluorescence measurement. However, the fluorescence intensity can easily vary due to changes in intensity fluctuation of excitation light, photobleaching, and light scattering [29]. In particular, the presence of metallic particles can alter the surrounding condition which may influence the optical properties of molecules. They also may act as collisional quenchers of fluorescence [30]. Moreover, we have to separately distinguish complexation-induced quenching from FRET-based fluorescence signal in a view of understanding for molecular dynamics. In general, the correlation between FRET efficiency and change in donor lifetime can be supported by below equation again,

                                                                                                                                                                                                                          (1)

where  and  are the fluorescence lifetime of the FRET donor in the presence and absence of FRET acceptor, respectively [31]. Since FRET efficiency is inversely proportional to fluorescence lifetime of donor fluorophore, thus the higher the FRET efficiency mean the shorter donor lifetime, suggesting decrease in excited lifetime of donor as great evidence of FRET. Observing the time-related fluorescence of FRET system will be helpful to optimize the condition for maximized FRET efficiency as well as to understand dynamic events involved in the intermolecular energy transfer phenomenon ~.

Point 2: The structure of FAM, TAMRA, CP should be clearly shown.

Response 2: We added the chemical structure of CPs, 6-FAM, and TAMRA to figure 1. And we revised manuscript and figure caption.

Figure 1 caption: ~ (c) The chemical structure of CPs, 6-FAM, and TAMRA.

In text: ~ The molecular structures of three fluorophores are described in Figure 1 (c).

Point 3: In the present form the text in the subsection 3. “Results and Discussion” is incomprehensible in fragments. The text should be improved and divided into more paragraphs and/or subsections.

Response 3: We revised the manuscript according to your comment. “Results and Discussion” is divided into subsection.

à Revised sentences (Page 3): 3. Results and Discussion

3.1. Two-step FRET process in CPs and two dyes-labeled aptamer complex

Two-step FRET system with three fluorophores have advantages compared to one-step FRET such as efficiency enhancement through relay station and better detection sensitivity due to lower background fluorescence of acceptor ~.

3.2. The spectral overlap between three fluorophores

Figure 2 (a), (b) and (c) show the intrinsic optical properties of three fluorophores (CPs, 6-FAM, and TAMRA), respectively ~.

3.3. The fluorescence decay dynamics of three fluorophores

For a better understanding of the energy transfer processes between FRET partners in the absence and presence of K+ ions, we measured time-resolved fluorescence ~.

3.3.1. The fluorescence decay of CPs

Figure 3 shows the fluorescence decay curve of CPs under 380 nm excitation where this wavelength corresponds to the absorption maximum of CPs and minimizes the direct absorption by ~.

3.3.2. The fluorescence decay of 6-FAM

The fluorescence decay dynamics of 6-FAM or TAMRA in complexation phase of CPs and aptamer was also measured ~.

3.3.3. The fluorescence decay of TAMRA

Finally, we measured the fluorescence decay dynamics of TAMRA in the absence and presence of K+ ions, respectively ~.

Point 4: The decay time of fluorescence components should be summarized in the form of table or added to the Figure 4.

Response 4: We summarized the fluorescence decay time in the form of table and revised the manuscript.

à Revised sentences (Page 12): In summary, Table 1 shows the fitted parameters for the fluorescence decay of CPs, 6-FAM, and TAMRA. The fluorescence decay time is calculated by single- or multi-exponential fitting as given below, [30]

, where  and  are the weight factors of each of the decay component.  and  are decay time. In the presence of additional decay pathways due to the conformational change or molecular interaction, the fluorescence decay time decreases and its curvature can be changed from single exponential to multi-exponential shape as well as intensity variation. Before and after complexation, this phenomenon representing occurrence of one or two-step FRET was commonly observed in CPs and 6-FAM. In case of TAMRA, the reduction in decay time may result in the change in surrounding environment. In the presence of K+ ions, two-step FRET process can be accelerated owing to shortened intermolecular distance and therefore fluorescence decay time more decreases.

Fluorophores

Free

w/o K+ 

with K+

(nm)

(ns)

(ns)

(ns)

(ns)

(ns)

CPs

380

0.43 (with K+)

0.94

0.35

0.06

1.76

0.73

0.19

0.27

1.46

6-FAM

380

-

-

0.49

-

-

-

0.33

-

-

490

3.72

0.51

0.36

0.49

2.09

0.61

0.25

0.39

2.18

TAMRA

380

1.01 (with CPs)

-

0.87

-

-

-

0.90

-

-

Point 5: A steady energy transfer efficiency was estimated as 230 %. The Authors should explain a result above 100%.

Response 5: We added some schematics to explain the steady energy transfer efficiency of 230%. In the absence of K+ ions, the population of TAMRA is supported by CPs through one-step FRET. Let us define the fluorescence intensity of TAMRA in the absence of K+ ions is 100 at 2 ns in arbitrary units. On the other hand, if K+ ions are involved, an additional two-step FRET occurs from CPs to TAMRA via 6-FAM. TAMRA plays an important role as an intermediator, where the energy transfer from CPs to TAMRA is mediated. Our energy transfer efficiency equation enables to estimate the transferred fluorescence from TAMRA in terms of the normalized ratio to the fluorescence intensity of TAMRA in the absence of K+ ions. When the increased fluorescence intensity of TAMRA at 2ns is 330, the energy transfer efficiency of 230% is obtained. It is surprising that two-step FRET results in a remarkable population enhancement. Such a large enhancement can be attributed to the delocalized electronic nature of CPs, giving rise to the energy harvesting effect.

à Revised sentences (Page 12):This equation was described in a view of gain or loss of exciton population during two-step FRET process. In the absence of K+ ions, the population of TAMRA is supported by CPs through one-step FRET. Let us define the fluorescence intensity of TAMRA in the absence of K+ ions is 100 at 2 ns in arbitrary units. On the other hand, if K+ ions are involved, an additional two-step FRET occurs from CPs to TAMRA via 6-FAM. TAMRA plays an important role as an intermediator, where the energy transfer from CPs to TAMRA is mediated. The energy transfer efficiency equation enables to estimate the transferred fluorescence from TAMRA in terms of the normalized ratio to the fluorescence intensity of TAMRA in the absence of K+ ions. When the increased fluorescence intensity of TAMRA at 2ns is 330, the energy transfer efficiency of 230% is obtained. It is surprising that two-step FRET results in a remarkable population enhancement. Such a large enhancement can be attributed to the delocalized electronic nature of CPs, giving rise to the energy harvesting effect ~.    

Point 6: Theoretical parts and discussion should be supported by references

Response 6: We added theoretical background and references related to fluorescence decay dynamics and revised manuscript for a better understanding.

à Revised sentences (Page 7): 3.3. The fluorescence decay dynamics of three fluorophores

For a better understanding of the energy transfer processes between FRET partners in the absence and presence of K+ ions, we measured time-resolved fluorescence. First of all, we measured the fluorescence decay of fluorophores before the complexation with two-dyes-labeled aptamer (denoted by ‘Free’) was measured individually to know intrinsic fluorescence decay time of fluorophores. Then, we compared it with the measured data in the absence and presence of K+ ions. The measured fluorescence decay curve was fitted by single- or multi-exponential decay function taking into account of the change in decay curvature [30]. In the fluorescence decay curve, the multiple decay components generally implies the contribution of additional decay pathways due to molecular interaction to whole fluorescence decay. Thus, we distinguished fast and slow decay components from multi-exponential decay curvature by taking into account of the intrinsic fluorescence decay time of fluorophores.

To calculate the energy transfer efficiency due to FRET, we used a rate equation or its time-dependent differential equation. In general, the fluorescence decay rate (k_F) is defined by the sum of all radiative (k_rad) and nonradiative decay (k_nonrad) components. [30]

                                                                                                                                               (3)

, where τ_rad and τ_nonrad represent radiative and nonradiative decay time, respectively. Since the fluorescence intensity (I) is proportional to the exciton population (N), the time-dependent fluorescence decay can be characterized by a differential equation form of exponential function.[39, 45]

                                                                                                                                                (4)

                                                                                                                                                  (5)

, where g indicates carrier (exciton) generation function by excitation light. τ_int represents an intrinsic fluorescence decay time including all radiative and nonradiative decay components. When decay pathways of FRET and charge transfer are involved, the rate equation can be modified by an additional nonradiative decay time ().[45]

                                                                                                                                                    (6)

In particular, this differential rate equation effectively provides the relation between an individual fluorescence decay rate and a time-dependent exciton population.

Point 7: Some editing issues:

- Paga 10 line 324

- Page 8 line 286

Response 7: We revised manuscript.

à , where and indicate the fluorescence intensity ~.

à , where  and  indicate the fluorescence intensities ~.

Reviewer 4 Report

Comment:

This article introduces a potassium ion detection assay by FRET method in conjugated polymer and dye-labeled aptamer. The authors displayed a series of data to verify the two-step FRET process with the addition of K+ in CPs/6-FAM/TAMRA system. There are some problems should be solved before considering the publication in this journal.

1. It hard to find a prominent difference among complexation and free CPs treated with K+ in Figure 3. It seems to clearer that assemble three divided figures into one figure.

2. It is essential to study a control experiment with different ratio of CPs and Dyes-aptamer in the absence/presence of K+. Because the concentration of participants has a concerned effect on FRET process.

3. It is interesting to study a reversible FRET process through adding K+ chelators such as crown ether in this system.

Author Response

Response to Reviewer 4 Comments

This article introduces a potassium ion detection assay by FRET method in conjugated polymer and dye-labeled aptamer. The authors displayed a series of data to verify the two-step FRET process with the addition of K+ in CPs/6-FAM/TAMRA system. There are some problems should be solved before considering the publication in this journal.

Point 1:It hard to find a prominent difference among complexation and free CPs treated with K+ in Figure 3. It seems to clearer that assemble three divided figures into one figure.

Response 1:

We modified Figure 3. Three divided figures assembled into one figure. And then we revised manuscript as shown in below figure.

Point 2: It is essential to study a control experiment with different ratio of CPs and Dyes-aptamer in the absence/presence of K+. Because the concentration of participants has a concerned effect on FRET process.

Response 2: In our previous research (Adv. Funct. Mater. 2014, 24, 1748–1757) of conjugated polymer and aptamer based potassium assay, we have investigated about detection sensitivity and the modulation of detection range. Without CPs, K+ ions (in a millimolar concentration range) were detected successfully with a limit of detection (LOD) of 22.5 μM ~ 3.3 mM. In the presence of CPs, highly sensitive detection was realized in a range of nanomolar K+ concentration with a LOD of ≈ 3 nM. The dynamic detection range was measured at different charge ratios of [+]:[-] =1:1 ~ 6:1. With increasing the charge, the CPs’ concentration around aptamers increases and more excitons can be transferred to 6-TAMRA with the intensified FRET-induced 6-TAMRA emission. This makes it possible to detect very low concentration of K+ ions. The modulation of the detection range and LOD is possible by controlling the signal amplification (or antenna) effect of CPs via simply adjusting the charge ratio.

Point 3: It is interesting to study a reversible FRET process through adding K+ chelators such as crown ether in this system.

Response 3: Thank you for your idea and comment. In our knowledge, members of the crown ether family have been widely applied in the design of fluorescence-based sensor systems. We also think that a reversible FRET is very useful in a view of research or application.

Round  2

Reviewer 2 Report

The changes made to the job have improved it. As previously mentioned, I do not reveal any inaccuracies or inconsistencies. However, in my opinion, the lack of novelty and the lack of relevance with the topic of the journal remains critical points.
Scientifically, the work is valid, so I refer the final decision to the editor.

Reviewer 3 Report

The manuscript entitled “Two-step Energy Transfer Dynamics in Conjugated Polymer and Dye-labeled Aptamer-based Potassium Ion Detection Assay” focuses on the time-resolved fluorescence measurement of the studied system. The revised manuscript has been significantly improved. The clarity of presentation has been improved. In my opinion the manuscript can be accepted for publication.